# Differential Role of Factor XIII in Acute Myocardial Infarction and Ischemic Stroke

**DOI:** 10.3390/biomedicines12030497

**Published:** 2024-02-22

**Authors:** Jan Traub, Martin S. Weber, Anna Frey

**Affiliations:** 1Department of Internal Medicine I, University Hospital Würzburg, 97080 Würzburg, Germany; frey_a@ukw.de; 2German Comprehensive Heart Failure Center, University and University Hospital Würzburg, 97080 Würzburg, Germany; 3Department of Neurology, University Hospital Göttingen, 37075 Göttingen, Germany; martin.weber@med.uni-goettingen.de; 4Fraunhofer Institute for Translational Medicine and Pharmacology ITMP, 37075 Göttingen, Germany

**Keywords:** factor XIII, fibrin stabilizing factor, acute myocardial infarction, ischemic stroke, mortality, biomarker, supplementation

## Abstract

Factor XIII is a transglutaminase enzyme that plays a crucial role in hemostasis and wound healing. It crosslinks fibrin strands, stabilizing clots and promoting clot resistance to fibrinolysis. Additionally, Factor XIII has been found to have multiple other functions that extend beyond coagulation, including the regulation of inflammation and tissue repair processes. Emerging evidence suggests that Factor XIII may also have differential roles in acute myocardial infarction and ischemic stroke, two common cardiovascular events with significant morbidity and mortality. In acute myocardial infarction, Factor XIII has been implicated in promoting clot stability and reducing the risk of re-occlusion. In ischemic stroke, Factor XIII may also contribute to the pathogenesis of cerebral ischemia by promoting clot formation and exacerbating neuronal damage. Several studies have investigated the association between Factor XIII and these cardiovascular events, using various approaches such as genetic polymorphism analysis, animal models, and clinical data analysis. These studies have provided important insights into the role of Factor XIII in acute myocardial infarction and ischemic stroke, highlighting its potential as a therapeutic target for interventions aimed at improving outcomes in these conditions. In this review, we will summarize the current understanding of Factor XIII’s role in acute myocardial infarction and ischemic stroke.

## 1. Introduction

Acute myocardial infarction (MI) and ischemic stroke (IS) are major causes of morbidity and mortality worldwide. They share similar risk factors, such as hypertension, smoking, and diabetes, and are characterized by the formation of thrombi, leading to ischemia in the heart or brain, respectively. Current therapeutic strategies for these conditions mainly focus on restoring blood flow to the affected area and preventing further clot formation. Here, improvements in heart and brain catheterization techniques allow for rapid reperfusion and clot removal. However, cardiac (i.e., development of heart failure) and neuronal (i.e., disability) morbidity call for further optimization in the treatment beyond the interventional options.

There is growing evidence that factors involved in the coagulation cascade, such as Factor XIII (FXIII), may play a significant role in the pathophysiology of these cardiovascular events and could potentially serve as therapeutic targets. In this review, we aim to sum up current knowledge on the role of FXIII in acute MI and IS. We will explore the different roles of FXIII in these two conditions and discuss the potential implications for therapeutic options.

## 2. Pathophysiology—Similarities and Differences

Both acute MI and IS are characterized by a sudden disruption to blood flow, leading to ischemia of the supplied area with necrosis and loss of function in the affected tissue. However, the etiology of both entities significantly varies, which is why the role of FXIII is important to have in mind during evaluation.

Acute MI is primarily caused by the rupture of unstable atherosclerotic plaques in the coronary arteries [1]. Thus, the release of thrombogenic factors and inflammatory cells leads to the formation of a thrombus that obstructs blood flow to the heart muscle. Necrosis of the myocardium is then caused by absolute coronary insufficiency. While acute complications include arrhythmias, sudden cardiac death, and myocardial ruptures, middle- and long-term complications involve the development of (chronic) heart failure, ventricular aneurysmata, embolic events (which can lead to IS), re-infarction, and pericarditis [2]. 

In contrast, the causes of IS are more diverse: it is often the result of cardiac emboli caused by atrial fibrillation, persistent foramen ovale, endocarditis, dilatative cardiomyopathy, or myxoma. Additionally, macro- and microangiopathy of the brain vessels can be the cause, for example, due to hypertension and diabetes mellitus. Less frequent causes are vasculitis, arterial dissection, or genetic disorders, such as cerebral autosomal-dominant arteriopathy with subcortical infarcts and leukoencephalopathy [3]. 

## 3. Coagulation Factor XIII

The identification of FXIII can be traced back to the 1940s when a ‘serum factor’ was discovered to cause fibrin clots to become insoluble in a concentrated urea solution. This factor was initially called ‘fibrin stabilizing factor’. A case study in 1960 provided evidence that a patient’s excessive bleeding resulted from a lack of fibrin stabilizing factor. Following this clinical observation, fibrin stabilizing factor became officially known as FXIII in 1963 [4]. 

As a pro-transglutaminase zymogen [5], FXIII circulates in the plasma in a tetrameric form. It consists of two potentially active, catalytic A subunits (FXIIIA) and two carrier/inhibitory B subunits (FXIIIB), while FXIIIB is in excess (Figure 1). Platelets and other cells of bone marrow origin mainly produce FXIIIA, while hepatocytes are responsible for FXIIIB secretion into the blood. The blood plasma contains heterotetramers at a concentration ranging from 14 to 28 mg/L, and their lifespan is between 9 and 14 days [6]. The activation of FXIII takes place on the fibrin surface within the plasma, with FXIIIA remaining bound to it and FXIII-B being released into the serum [6,7]. FXIIIA inherits a transamidase function that inserts ε-(γ-glutamyl)lysyl isopeptide connections into protein targets. These links can develop within a single target, like fibrin, or between distinct proteins (i.e., alpha2-plasmin inhibitor to the fibrin alpha chain) and may influence their biological role. Thus, FXIIIA is involved in the final stages of clot formation to stabilize clots and promote resistance to fibrinolysis [8]. Summing up, plasmatic FXIII plays a crucial role in maintaining the integrity and stability of blood clots, which underscores its relevance in both MI and IS.

Furthermore, the dimer of FXIIIA, but not FXIIIB, is present within platelets, monocytes, macrophages, chondrocytes, osteoblasts, and osteocytes [9]. Intracellular FXIIIA is thought to be involved in platelet adhesion, retraction, and monocyte/macrophage phagocytosis [10]. Platelets can absorb a small quantity of FXIII from the plasma, which is found in α-granules. However, this quantity is insignificant when compared to the amount of FXIIIA present in the cytoplasm [11]. The estimated FXIIIA content in a single platelet is around 60 ± 10 fg, representing approximately 3% of the total platelet proteins. This results in a concentration of FXIIIA that is approximately 100 to 50 times higher in platelet cytoplasm compared to plasma [12]. Elevation in the intracellular levels of Ca^2+^ due to thrombin activation leads to the initiation of cellular FXIII activation within the cell, independent of proteolysis [13].

It is very important to understand that FXIII is a multifunctional enzyme with important intra- and extracellular functions beyond the coagulation cascade, which are attributed to the transglutaminase activity of the FXIIIA subunit [14]. First of all, extracellular collagens, fibronectin, fibrinogen, vitronectin, osteopontin, and osteonectin are major substrates of FXIIIA, which are involved in crosslinkage and stabilization in various conditions and organs [15]. In wound healing, FXIII induces crosslinking of the provisional matrix and influences the interaction of leukocytes with the provisional extracellular matrix. Furthermore, FXIII impacts endothelial cell vascular epithelial growth factor-2 and αvβ3 integrin, resulting in a reduction in antiangiogenic protein thrombospondin-1. This encourages angiogenesis and the formation of new blood vessels [16]. Finally, in cases of infectious diseases, FXIII binds bacterial surface proteins to fibrinogen through crosslinking, leading to the immobilization and destruction of the bacteria [16].

In light of acute thrombotic events like acute MI and IS, it is essential to keep in mind that FXIII is more than just an intravascular coagulation factor [4]. Especially when it comes to wound healing and the repair of injured heart or brain tissue, the transglutaminase activity of FXIII is essentially involved in the crosslinking of extracellular structural proteins, angiogenesis, and the regulation of inflammation.

## 4. Factor XIII Deficiencies—Congenital vs. Acquired

Congenital FXIII deficiency is an uncommon hereditary bleeding disorder inherited in an autosomal recessive manner [17]. In its most extreme manifestation, it may manifest as uncontrolled bleeding in the early stages of infancy. This can be attributed to a genetic mutation in either FXIIIA (type 2) or FXIIIB (type 1) genes, with the majority (95%) of FXIII-related bleeding disorders being linked to an FXIIIA deficit [18]. The gene encoding the A subunit is situated on chromosome 6p24-p25 and consists of 15 exons and 14 introns. Up to now, well over 150 genetic alterations of FXIIIA have been documented, with around fifty percent being missense mutations [19]. The B subunit gene is situated on chromosome 1q31-32.1 and consists of 12 exons and 11 introns. A total of 16 mutations have been documented, predominantly in the form of missense mutations [20]. The B-subunit serves as a transporter for subunit A. Consequently, the absence or malfunction of subunit B undermines the stability of the heterodimeric FXIII complex and results in a reduced level of FXIIIA. As a result, individuals with subunit B deficiency exhibit less severe bleeding symptoms and experience a decline in FXIII activity to approximately 5% to 10% [21]. 

Acquired FXIII deficiency is more common and can result from either excessive consumption (such as through surgery, disseminated intravascular coagulation, inflammatory bowel disease, Henoch–Schönlein purpura, sepsis, or thrombosis), reduced production (due to conditions like liver disease, leukemia, or medications such as valproic acid and tocilizumab), or immune-mediated processes (including systemic lupus erythematosus, rheumatoid arthritis, hematologic malignancy, and certain medications like isoniazid) [17]. It is crucial to identify acquired FXIII deficiency and its root cause, as treatment choices differ based on the cause [22]. Acquired FXIII deficiency is significantly linked to high rates of illness and death among critically ill individuals, regardless of the underlying cause [23].

## 5. Plasma vs. Tissue Quantification of FXIII Activity

Traditionally, a plasma clot soluble in concentrated urea solution or in dilute acids was considered to be a sign of FXIII deficiency [24]. The quantification of plasmatic FXIII activity, which is nowadays routinely used in various clinical settings, is based on two assay principles [25]: on the one hand, ammonia released during the transglutaminase reaction is measured; on the other hand, labeled amine incorporated into a protein substrate is quantified. In both assays, thrombin and Ca^2+^ are used to activate FXIII. The results for FXIII activity are normally given in % of normal.

As described above, FXIII also appears to be crucial in wound healing and tissue repair. In an experimental murine setting, tissue FXIII activity was imaged in vivo via single-photon emission computed tomography [26]. The researchers utilized a peptide with an ^111^Indium label for affinity (^111^In-DOTA-FXIII) [27,28]. FXIII identifies the probe as a substrate and links it to proteins in the extracellular matrix, resulting in the localized immobilization of ^111^In-DOTA-FXIII [29]. Interestingly, intravenous FXIII pro-enzyme supplementation increased ^111^In-DOTA-FXIII uptake, suggesting increased tissue FXIII activity. Researchers also successfully used a fluorine-18-labeled positron emission tomography agent and others to monitor tissue FXIII activity non-invasively [30,31].

## 6. Factor XIII in Myocardial Infarction

The query ((Factor XIII) OR (FXIII)) AND ((myocardial infarction) OR (heart attack)) was used for the systematic literature review of the electronic databases Embase, PubMed, and Web of Science on 4 January 2024. A total of 144 original articles and reviews could be identified between the years 1971 and 2024. From this search, 83 (58%) of these articles were suitable for this review section, while 61 excluded articles did not cover the area of interest, were not available, or were just short reports or commentaries.

Briefly, FXIII is one of about 500 clot-bound proteins that have been identified within plasma fibrin clots in MI [32]. A study conducted in 1987 indicated increased activity of both thrombin and FXIII activity in patients with acute MI [33]. Besides its role in the coagulation cascade, it also appears crucial for wound healing and tissue repair, in particular after MI [34]. 

### 6.1. FXIII and Risk of Myocardial Infarction

In 119 young (median age 36 years) MI patients, FXIII activity (i.e., above 75 percentile value, 127%) increased the risk of MI in the unadjusted model significantly; however, after adjustment for gender, hyperlipidemia, and smoking, the association was not significant [35]. A different study involving 955 participants revealed that higher FXIII levels were linked to a notably elevated risk of myocardial infarction in women but not in men [36]. In individuals with confirmed coronary artery disease who undergo percutaneous coronary intervention, elevated FXIII activity was linked to a slight rise in cardiovascular risk, characterized by the recurrence of MI or mortality [37].

In another study, 63 male participants without clinical signs of coronary artery disease who subsequently experienced a myocardial infarction exhibited lower adjusted FXIII A-subunit levels at the time of recruitment (129.2% vs. 113.3%) compared to 164 control individuals [38]. The authors explained that lower levels of FXIII A-subunit antigen in the laboratory were associated with higher thrombin generation, leading to an increased risk of experiencing blood clotting events [38]. 

Further, FXIII B-subunit levels were reported to positively relate to risk factors for cardiovascular disease, suggesting an underlying association with insulin resistance [39].

Interestingly, congenital FXIII deficiency was unable to protect against acute coronary diseases in a case series [40]. Additionally, venous thrombosis occurred in a patient with FXIII deficiency [41]. 

### 6.2. FXIII Dynamics in Myocardial Infarction

A transient drop in mean FXIIIA levels was noted in a cohort of 350 acute MI patients within the first five days after the event [42]. The FXIIIA drop was independent of the amount of injury assessed when using troponin and creatinine kinase levels. Our group recently confirmed that plasmatic FXIII activity after MI was highly dynamic, exhibiting a significant decline in the early healing period, with reconstitution 6 months later in a cohort of 146 MI patients [43]. Another study of 119 young (median age 36 years) MI patients even claimed that three months after MI, plasmatic FXIII activity was significantly higher in patients with MI compared to matched controls [35].

An inverse correlation between FXIII and the percentage of high-molecular-weight fibrin(ogen) complexes was demonstrated early after MI in an old investigation from 1977 [44]. Back then, the authors and others [45] suggested that the depression of FXIII during MI is secondary to extravascular or intravascular coagulation and may reflect its degree [44]. In a previous study, most patients exhibited positive ethanol gelation tests after the first day, peaking at day 5. This coincided with peak factor VIII activities and showed a negative correlation to FXIII activity when measured biologically [45].

### 6.3. Prognostic Significance of FXIII after Myocardial Infarction

Depressed FXIII activity early after MI predicted a greater size of MI (assessed in magnetic resonance imaging) and lower left ventricular ejection fraction after 1 year [43]. Equally, in another cohort, those who developed post-MI heart failure showed the highest drop of FXIIIA levels in the days after MI [42], and, importantly, they already presented with low levels at recruitment. In a murine study with FXIII knockout mice, multichannel fluorescent molecular tomography revealed impaired healing after MI compared to controls [46].

Besides the development of heart failure, ventricular ruptures after MI were linked to disturbed FXIII activity. Nahrendorf et al. showed that murine FXIII deficiency causes cardiac rupture, impairs wound healing, and aggravates cardiac remodeling in mice with MI [47]. Interestingly, skin healing was impaired in the same mouse model [48]. A superior quality of the cardiac matrix is a requisite for the maintenance of structural and functional integrity after MI [34]. In this line, immunoblotting revealed lower FXIII levels in the myocardium of nine patients with infarct rupture when compared to MI patients without rupture [26].

Another report relating to 333 acute MI patients found that FXIIIA levels assessed at d0 and d4 were independent predictors of major adverse cardiovascular events in long-term follow-up [49]. Interestingly, FXIIIA levels at day 4 showed the strongest association, suggesting that the protective role of FXIIIA is maximized when high levels are maintained for a longer time [49].

Among 257 serum proteins, which were quantified from the peripheral blood of 229 acute MI patients, only increased levels of actin beta-like 2 and FXIIIA were significantly associated with ventricular fibrillation before guided catheter insertion for primary percutaneous coronary intervention [50]. 

Regarding death, the survival analysis ascribed an increased probability of early death inversely related to FXIII-A quartiles [42]. Cardiogenic shock and ventricular rupture were death causes in three patients with low FXIII activity (57 ± 8%) after MI [51].

### 6.4. Val34Leu FXIII Polymorphism in Myocardial Infarction

Multiple single-nucleotide polymorphisms in the FXIIIA subunit gene have been assessed, revealing that Val34Leu is the primary functional polymorphism affecting FXIII activation. Additionally, FXIII Val34Leu represents the sole prevalent polymorphism within the coding region of the A-subunit gene of FXIII linked to coronary artery disease [52]. Methods for evaluating the FXIII-A Val34Leu polymorphism are currently time-consuming and labor-intensive. Recently, there has been a simplification in detection through the use of real-time polymerase chain reaction with fluorescence resonance energy transfer detection and melting curve analysis [53]. Differential peptide display offers a straightforward and user-friendly method for phenotype assessment, with benefits over PCR-based assays due to its quicker speed and direct analysis of the specific compound under investigation [54]. The FXIII Val34Leu mutation is a frequently observed genetic variation that has been linked to faster blood clot formation and the decreased breakdown of clots [55]. The Val34Leu polymorphism is situated close to the thrombin activation site of FXIII and leads to a more significant and quicker susceptibility of FXIII to thrombin activation. However, this genetic polymorphism does not impact the specific enzyme activity of FXIII [56]. It has been proposed that the substantial heritability of FXIII activity can likely be mostly accounted for by the significant impact of the Val34Leu polymorphism on FXIII activity [57]. During the thrombin-induced activation of FXIII, the Val34Leu mutation leads to the more rapid development of dimerization in the fibrin gamma chain and polymerization in the alpha chain [56]. Some have proposed that the Val34Leu mutation is linked to higher FXIII activity in Asian Indian women while asserting that both genetic and racial factors are important for determining plasma FXIII levels [58]. Increased consumption of FXIII may potentially account for the reduced FXIII levels in individuals with homozygous Leu34 coronary artery disease [59]. Interestingly, fibrin clots formed in the presence of V34L polymorphism tend to be thinner and less porous [60]. In 242 young individuals with MI, patients produced stiffer fibrin clots and displayed reduced response to fibrinolysis with a lower fibrinolysis rate than healthy controls [61]. Another report of 137 twin pairs indicated that genetic and environmental influences are important in determining fibrin clot structure and function [62]. The association between FXIIIA Val34Leu polymorphism and MI risk is controversial, with conflicting results in a large amount of the studies conducted, which is probably due to the multigenic nature of MI, whereby single polymorphisms are only bound to play, at best, a limited role in the global risk of disease [63]. 

Briefly, meta-analyses have concluded that FXIII Val34Leu polymorphism is associated with increased risk for coronary artery disease, especially MI, but not with coronary artery disease without MI [64,65]. In contrast, other meta-analyses have suggested that FXIIIA Val34Leu polymorphism is a protective factor for MI generally [66] and in Caucasians [67], which was also concluded by recent review articles on this controversial issue [68,69]. Two comprehensive meta-analyses have suggested that being a carrier of the Val34Leu allele and FXIII heterozygotes themselves are associated with a protective effect against premature MI [70]. Another meta-analysis claimed that the protective effect of the FXIII-A Val34Leu polymorphism could be significantly influenced by interactions between genes and environmental factors [71].

In this line, the protective effect of Val34Leu becomes stronger as fibrinogen concentration increases, potentially explaining these contradictory results [72]. It appears that the combined effects of multiple variations, both in terms of their interaction and their impact on various traits, play a role in determining the levels of plasma fibrinogen, the structure of fibrin clots, and the risk of myocardial infarction [73]. The interplay of the genetic makeup and environmental factors in determining the appropriate plasma level offers a potential rationale for the varying reactions of individuals to their surroundings [74]. During the thrombin-induced activation of FXIII, the Val34Leu mutant exhibited accelerated development of fibrin gamma-chain dimerization and alpha-chain polymerization [56]. The elevated speed of fibrin stabilization due to the Val34Leu FXIII appears to be contradictorily linked to a defensive impact against abnormal blood clot formation [56]. The Val34Leu allele may also influence the prothrombotic state by modulating the inflammatory state, as it was associated with increased interleukin-6 levels [75]. Investigations have also been conducted on an artificially designed Val34Leu peptide, which takes on a notably distinct structure. In this conformation, the larger L34 occupies the nonpolar binding site while the V29 side chain is exposed to the surrounding solvent. This arrangement may accelerate the release of FXIII from thrombin following its activation [76]. Deciphering the intricate connections between environmental factors and genetic components will be a significant challenge, but understanding prothrombotic elements and their diverse genetic traits may uncover a shared mechanism through which several of these risk factors function [77].

Regarding outcomes after the occurrence of acute MI, carriers of Val34Leu were reported to have a lower risk of developing ischemic heart disease after MI [49] and exhibited improved survival after MI [78]. The presence of the FXIII L34 allele demonstrated enhanced survival following myocardial infarction across all of the studied groups, possibly due to its increased activity linked to potential positive impacts on healing of the heart muscle and restoration of functions in 416 MI patients [78]. However, FXIII and fibrinogen variations were not linked to the occurrence of sudden death from ventricular fibrillation during myocardial infarction [79].

### 6.5. Role of His95Arg FXIII Polymorphism in Myocardial Infarction

An A→G transition in the 95th codon of the gene responsible for producing subunit B of FXIII has been discovered, resulting in the substitution of histidine with arginine. This His95Arg variation is quite prevalent and has been linked to an accelerated dissociation rate of the FXIII A2B2 tetramer after activation by thrombin [80].

The presence of the FXIIIB Arg95 allele seems to have little association with MI risk [81]. Another research indicated that the risk of myocardial infarction for individuals with one copy of the Arg95 allele of the FXIIIB His95Arg polymorphism is not reduced, unlike those with two copies, where the risk appears to be decreased by 0.3-fold [82].

### 6.6. Diagnostic Considerations

Molecular techniques are quickly uncovering the molecular and cellular mechanisms involved in MI [83]. Molecular imaging of cardiac FXIII activity in vivo could potentially allow us to gain insight into the heart and obtain valuable data on scar development and associated prognosis following myocardial infarction [34].

### 6.7. Therapeutic Considerations

The matricellular proteins thrombospondins, osteopontin, and osteoprotegerin, along with FXIII, as a transglutaminase, could offer new treatment opportunities to enhance the development and reinforcement of the infarcted heart’s matrix in the initial days following MI [34]. Asani et al. [49] discussed the existence of an early specific time window in which to operate to maintain optimal FXIIIA levels and sustain heart repair. In patients undergoing cardiac surgery involving cardiopulmonary bypass, a FXIII repletion study aiming to prevent postoperative bleeding found dosages of 35 IU/kg recombinant FXIII-A to be an adequate dosage [84]. Importantly, all of the 18 treated patients (except one with MI) recovered without sequelae and without other thromboembolic events [84], suggesting that FXIII repletion might also be safe in MI patients. 

Following myocardial infarction, the enlargement of the left ventricle was reduced in mice treated with FXIII compared to those treated with saline [26]. Quantitative histological and reverse transcription–polymerase chain reaction evaluations demonstrated that FXIII therapy expedited the resolution of the neutrophil response, improved macrophage recruitment, raised collagen levels, and boosted angiogenesis within the healing infarct. In a separate study involving the targeted blocking of mineralocorticoid receptors immediately after myocardial infarction in rats, there was an observed increase in macrophage infiltration and temporary upregulation of healing-promoting cytokines and FXIII expression within the damaged heart tissue. This led to improved neovascularization of the infarct area as well as reduced early left ventricular dilation and dysfunction [85].

Recently, advanced treatments based on FXIII have been suggested for counteracting adverse effects following myocardial infarction. This includes the proposal for intra-myocardial injection of biomaterial that can be modified by FXIII [86]. In a porcine investigation, targeted myocardial microinjection of a biocomposite (containing fibrinogen, fibronectin, FXIII, gelatin-grafted alginate, and thrombin) attenuated the post-MI decrease in left ventricular wall thickness and infarct expansion [87]. Nahrendorf’s group determined that the documented clinical observations and corresponding experimental evidence provide backing for the idea that a shortage of FXIII may hinder recovery in individuals with MI, whereas treatment replacement could help ease complications [52]. A future investigation is needed to establish the medical importance of these results:

It should be noted that the optimal threshold for defining post-MI acquired FXIII deficiency remains unclear. There is currently a lack of comprehensive observational studies measuring plasma FXIII activity shortly after MI and linking it to outcomes such as ischemic heart failure, re-hospitalization, or mortality. It is worth mentioning that FXIII activity assays are readily available in clinical practice, which could streamline the collection of these crucial data. Subsequently, initial human pilot studies or case reports can help assess the safety and appropriate dosage of intravenous FXIII replacement therapy. The existing use of FXIII concentrates (i.e., fibrogammin) in routine clinical settings (e.g., before surgery in patients with FXIII deficiency) provides a significant advantage in this context. Finally, randomized clinical trials will be necessary to determine if early supplementation with FXIII after MI could serve as a viable therapeutic option for patients with (acquired) FXIII deficiency.

## 7. Factor XIII in Ischemic Stroke

The query ((Factor XIII) OR (FXIII)) AND (ischemic stroke) was used for a systematic literature review of the electronic databases Embase, PubMed, and Web of Science on 10 January 2024. A total of 66 original articles and reviews could be identified between the years 1973 and 2024. In this study, 41 (62%) of these articles were suitable for this review section, while 25 excluded articles did not cover the area of interest, were not available, or were just short reports or commentaries.

### 7.1. FXIII in Ischemic Stroke

The involvement of FXIII in atherogenesis and its role in atherothrombotic IS has been discussed for years [88]. Briefly, studies suggest a key role of FXIII in thrombus stabilization, α2-antiplasmin crosslinking, and lysis resistance [89]. Platelet FXIII also controls the adhesive function of integrin to prevent excessive platelet buildup on extremely reactive thrombotic surfaces [90]. Reanalyzing the proteomes of human thrombi associated with cardioembolic and atherothrombotic cerebrovascular events and employing an artificial intelligence algorithm to investigate protein patterns revealed a marked association of FXIII with cardioembolic thrombi [91]. In a study analyzing the proteins in serum samples from 20 individuals with IS, it was found that ceruloplasmin, alpha-1-antitrypsin, von Willebrand factor, and coagulation FXIII B chain were able to distinguish IS patients from healthy control subjects [92]. The crosslinkage of α2-antiplasmin to fibrin by activated factor XIII is crucial for limiting fibrinolysis. Higher FXIII levels led to an increased binding of α2-PI to fibrin clots [93]. Recombinant tissue plasminogen activator, which is intended to target fibrin, leads to a reduction in circulating fibrinogen levels and is closely associated with a decrease in plasminogen. Additionally, there is an observed simultaneous rise in anti-fibrinolytic factors such as FXIII and alpha2-antiplasmin [94].

### 7.2. Genetic Variants of FXIII in Ischemic Stroke

Factor XIII should be regarded as an additional coagulation factor that plays a role in the intricate interplay between genetic and environmental factors, which is significant in the development of cardiovascular, cerebrovascular, and thromboembolic conditions [38]. Genetic variants like single-nucleotide polymorphisms rs5985 were associated with FXIII activity in the EuroCLOT study [95]. In a Sardinian sample with 294 patients, FXIII G185T did not contribute to IS predisposition either in crude or adjusted analyses [96]. A large meta-analysis of 25 studies involving 6100 IS patients and 9249 healthy controls concluded that FXIIIA rs5982 and rs3024477 polymorphisms are not associated with IS risk [97]. 

The FXIIIA Tyr204Phe allele was strongly associated with IS in a population-based case–control study in 190 women aged 20 to 49 years and 767 controls [98]. Another case–control study with 220 patients did not confirm the previously reported association between the FXIII Tyr204Phe variant and IS [99]. 

### 7.3. Val34Leu Polymorphism

Testing multiple polymorphisms in the FXIIIA subunit gene 201 in dizygotic twin pairs indicated that Val34Leu is the main functional polymorphism influencing FXIII activation [100]. In a study including 316 patients who died from athero-thrombotic IS, the FXIII-A Val34Leu polymorphism did not influence the occurrence of IS but had an effect on the severity of its outcome [101]. Other meta-analyses confirm that the Val34Leu polymorphism provides moderate protection against venous thromboembolism but not against IS; gene-gene and gene-environmental interactions might modify its effect [102]. In line, FXIII Val34Leu polymorphism failed to influence the risk of IS 496 patients and 1146 controls [103]. In another report with 599 patients and 100 controls, Val34Leu was not associated with increased IS risk as well [104]. Haplotype 11 of FXIIIa1 was linked to a higher risk of IS and contains minor alleles for Val34Leu and P565L. However, neither amino acid substitution showed a significant association with an increased incidence of IS among 368 IS patients [105]. A meta-analysis of 8800 individuals found no conclusive evidence linking the FXIII-A Val34Leu polymorphism with IS [106]. In a different meta-analysis encompassing 627 cases and 1639 controls within the young adult population, no significant association was observed between the FXIII Val34Leu polymorphism and undetermined source IS in any of the genetic models analyzed [70]. In this line, others concluded that screening for the FXIII Val34Leu polymorphism would not contribute significantly to the risk prediction of cerebrovascular disease [68]. In line, a study with 167 IS patients study suggests that the analysis of FXIII Val34Leu is not a useful diagnostic procedure in the work-up of IS [107]. Another meta-analysis of all candidate gene association studies in IS included data from 120 case–control studies and concluded that there was no statistically significant association of FXIII with IS [108]. Finally, a Greece case–control study found no differences in FXIII-AVal34leu allele frequencies between 51 IS patients and 70 controls [109].

In contrast, FXIII V34L was significantly more frequent in patients with first IS than in the control group in an Albanian population [110]. Another study suggests that in young women, the increase in IS risk conveyed by prothrombotic factors, including FXIII, is overall higher than that in MI [111]. According to an Indian study, FXIII V34L may be a significant risk factor for cryptogenic IS in the young [112]. Others showed a significant stroke risk interaction between the F13A1 polymorphisms rs2154299 and rs12194855 with estrogen plus progestin (E+P) treatment for IS [113].

### 7.4. FXIII Polymorphisms and FXIIIB

Interestingly, genetic markers associated with low FXIIIB levels increase the risk of the IScardioembolic subtype [114]. A South Asian investigation found an association between FXIII B-subunit antigen levels and a family history of stroke [115]. Further, the FXIII subunit B gen His95Arg variant was more common in IS patients with cerebral large vessel disease [116].

### 7.5. FXIII Polymorphisms and Pediatric Strokes

Regarding pediatric strokes, a study with 38 participants found a 2.21-fold increased risk of childhood IS in FXIII-A Leu34 allele carriers [117]. This was confirmed by an analysis with 282 IS patients and 430 controls, where FXIII Val34Leu allele frequency was significantly higher in patients with IS than in the controls [118]. In a study including 33 children, the presence of more than five polymorphisms (including FXIII Val34Leu) was associated with a higher risk for IS occurrence in children [119]. These findings could not be confirmed in a study group consisting of 392 individuals, including 81 children with IS, their biological parents (n = 162), and 149 control children [120]. Another report with 90 children concluded that, except for factor V Leiden mutation, no definite conclusion could be reached regarding the involvement of the studied mutations/polymorphisms (including FXIII) in childhood IS [121]. In this line, a recent meta-analysis with 358 children with stroke and 451 controls demonstrated a lack of association between FXIII polymorphisms and childhood IS [122].

### 7.6. FXIII Polymorphisms and Outcome

An observational research study discovered that there is a connection between the FXIII Val34Leu genetic variation and reduced blood clot levels, leading to improved functional results in individuals with acute IS who are treated with intravenous thrombolysis [123]. The FXIII Val34Leu polymorphism is linked to a higher mortality rate, with 20.0% of L34 carriers and 9.1% of patients with the V/V genotype dying after receiving thrombolytic therapy in a study involving 200 IS patients [124]. The harmful impact of this mutation appeared to be worsened by elevated fibrinogen levels, emphasizing the influence of fibrinogen levels on the hemostatic outcomes linked to the FXIII gene variation [124].

### 7.7. FXIII Levels and Outcome

Based on a study involving 69 IS patients, it was found that the level of plasma FXIII may have significance as an independent factor in predicting the risk of post-thrombolytic bleeding in IS patients [125]. This was not supported by a study with 114 patients, where none of the hemostatic markers analyzed (including FXIII) in the study predicted symptomatic cerebral hemorrhage in patients with IS treated with intravenous tissue plasminogen activator [126]. Similar conclusions were drawn in a study with 63 patients, where only alpha2-antiplasmin, but not FXIII, was predictive of successful recanalization [127]. An 8-month-old infant also had a case of hemorrhagic stroke due to severe FXIII deficiency. Rapid whole-genome sequencing revealed a new variant in the F13A1 gene, c.1352_1353delAT (p.His451ArgfsTer29) [128]. In a study involving 132 IS patients, it was found that low FXIII levels following intravenous thrombolysis are associated with short-term mortality. Analysis using a multiple logistic regression model showed that having FXIII levels in the lowest quartile 24 h post-lysis independently predicted mortality within 14 days after the event [129]. 

### 7.8. Imaging of FXIII in Ischemic Stroke

Near-infrared fluorescence imaging in IS mice revealed the spread of the FXIIIa-targeted probe in the area affected by ischemia and adjacent micro-vessels. The signals from the FXIIIa-targeted probe exhibited a strong correlation with images of immune-fluorescent fibrin staining [130]. This method could offer a practical experimental approach to studying the function of fibrin in stroke using live organisms [131].

### 7.9. Therapeutic Implications

In a mouse model of IS, administering FXIII inhibition prior to stroke induction reduced the size of the infarct, crosslinking of α2AP, and local microthrombosis. This led to improvements in motor coordination and fibrinolysis without causing intracranial bleeds [89]. Post-stroke FXIII inhibition also reduced brain injury and neurological impairments [89].

In a rat study, there was a notable intensification of focal neuronal necrosis and degeneration in the central area, while greater inflammatory infiltration and FXIII expression were observed in the surrounding region. This indicates heightened transglutaminase activity in these particular areas [131]. Nahrendorf et al. observed an irregular activation of the coagulation cascade and heightened inflammation (myeloperoxidase activity) in the vicinity where emboli are located in the brains of mice with infarction [132].

## 8. Conclusions

The differential role of FXIII in the athero-thrombotic conditions of MI and IS seems to be more complex and ambivalent than previously thought. On the one hand, heightened activity would promote the maintenance of fibrin buildup and greater plaque burden. Conversely, it would also decrease plaque susceptibility and the likelihood of embolism downstream [133]. Further, especially in MI, increased FXIII tissue activity in the infarcted tissue seems to provide beneficial effects by stabilizing the injured tissue. 

As shown above, the focus of conventional FXIII research has so far been on the role of FXIII polymorphisms and MI or IS risk, especially Val34Leu. While the Val34Leu variant seems to be protective for MI in the young, IS risk appears to be unaffected by this polymorphism. These conflicting results may explain why genetic testing has not been established to quantify MI/IS risk so far.

Most interestingly, low plasma FXIII activity directly after MI or IS (so-called acquired FXIII deficiency) predicts both short-term morbidity (i.e., cardiac rupture or intracranial hemorrhages) and mortality in both conditions (Figure 1). While the consumption of FXIII due to thromboembolism explains a transient drop in FXIII, very low FXIII activity (i.e., <60%) seems to impede both the vascular (i.e., bleeding) and extravascular (tissue repair) functions of FXIII.

Until now, except for promising murine MI studies, there is little evidence of whether FXIII supplementation after an index event may be beneficial in both conditions. Thus, a stronger focus of FXIII research on the extravascular transglutaminase properties may pave the way towards therapeutic applications of clinically already available FXIII (i.e., fibrogammin^®^), which has not been assessed in this context so far. Here, FXIII might be able to prevent early complications and increase short-term survival in MI and IS patients with low FXIII levels.

## Figures and Tables

**Figure 1 biomedicines-12-00497-f001:**
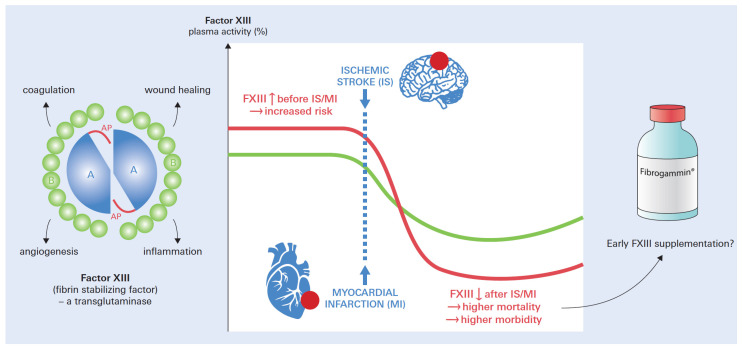
Structure, functions, and role of coagulation factor XIII in ischemic stroke and myocardial infarction. Factor XIII is an unusual blood coagulation factor, circulating as a heterotetramer composed of two catalytic A-subunits (A), two non-catalytic B-subunits (B), and activation peptides (APs). It is a member of the transglutaminase family with multiple functions, which extend its important role in the final stages of the coagulation cascade. Increased (↑) FXIII activity has been associated with an increased risk for ischemic stroke and myocardial infarction. In turn, low (↓) FXIII activity after an index event (acquired FXIII deficiency) relates to increased morbidity and mortality in both conditions. More research is needed to find out if and how FXIII supplementation may be beneficial in this regard.

## Data Availability

No new data were created or analyzed in this study. Data sharing is not applicable to this article.

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
