# Peer review of "Differential Role of Factor XIII in Acute Myocardial Infarction and Ischemic Stroke"

_biomedicines, 2024, doi:10.3390/biomedicines12030497_

Round 1

Reviewer 1 Report

Comments and Suggestions for Authors

This is a comprehensive review on the role of FXIII level and FXIII-A and FXIII-B subunit polymorphisms in acute myocardial infarction and ischemic stroke.

The paper is generally well written, and I agree with the conclusions. I just have a few comments:

1.     The concentration of the heterotetramer in the plasma is 14-28 mg/L (line 77)

2.     “it crosslinks fibrin strands by forming isopeptide bonds between various glutamines and lysines of the fibrins” (lines 80-82)

This statement is too general, it should be clarified, since we know exactly which fibrin chains and which lysine and glutamine side chains are involved in the formation of cross-links. The crosslinking of alpha2-plasmin inhibitor to the fibrin alpha chain should also be mentioned.

3.     Lines 117-120: References regarding the number of reported FXIII mutations are quite old; the databases now contain well over 153 FXIII mutations. These data should be updated, e.g. based on article of Javed H. (Javed H, Singh S, Ramaraje Urs SU, Oldenburg J, Biswas A. Genetic landscape in coagulation factor XIII associated defects - Advances in coagulation and beyond. Blood Rev. 2023 May;59:101032. doi: 10.1016/j.blre.2022.101032.)

Comments on the Quality of English Language

The manuscript contains some typos, word repetitions and confusing sentences.

Author Response

This is a comprehensive review on the role of FXIII level and FXIII-A and FXIII-B subunit polymorphisms in acute myocardial infarction and ischemic stroke. The paper is generally well written, and I agree with the conclusions. I just have a few comments.

             Thanks for this positive and constructive feedback.

  1. The concentration of the heterotetramer in the plasma is 14-28 mg/L (line 77)

This is a very important remark. We corrected the plasma concentration accordingly on page 2 of the revised manuscript:

“The blood plasma contains heterotetramers at a concentration ranging from 14 to 28 mg/L, and their lifespan is between 9 and 14 days [6].”

  1. “it crosslinks fibrin strands by forming isopeptide bonds between various glutamines and lysines of the fibrins” (lines 80-82)

This statement is too general, it should be clarified, since we know exactly which fibrin chains and which lysine and glutamine side chains are involved in the formation of cross-links. The crosslinking of alpha2-plasmin inhibitor to the fibrin alpha chain should also be mentioned.

We apologize for this very general statement. We clarified it accordingly on page 2 of the submitted manuscript:

“FXIIIA inherits a transamidase function that inserts ε-(γ-glutamyl)lysyl isopeptide connections into protein targets. These links can develop within a single target, like fibrin, or between distinct proteins (i.e. alpha2-plasmin inhibitor to the fibrin alpha chain) and may influence their biological role.”

  1. Lines 117-120: References regarding the number of reported FXIII mutations are quite old; the databases now contain well over 153 FXIII mutations. These data should be updated, e.g. based on article of Javed H. (Javed H, Singh S, Ramaraje Urs SU, Oldenburg J, Biswas A. Genetic landscape in coagulation factor XIII associated defects - Advances in coagulation and beyond. Blood Rev. 2023 May;59:101032. doi: 10.1016/j.blre.2022.101032.)

Indeed, the reported numbers came from quite old publications. We apologize for this inconsistency and included the updated numbers as well as the accurate, recent publication (PMID: 36372609) to the revised manuscript on page 3:

“The gene encoding the A subunit is situated on chromosome 6p24-p25 and consists of 15 exons and 14 introns. Up to now, well over 150 genetic alterations of FXIIIA have been documented, with around fifty percent being missense mutations [19].”

Reviewer 2 Report

Comments and Suggestions for Authors

The manuscript by Jan Traub and co-authors provides a comprehensive summary of the current understanding on the role of Factor XIII in acute myocardial infarction and ischemic stroke. This is based on information generated by clinical studies, animal models and genetic polymorphism analysis, which overall have revealed a differential role of Factor XIII in acute myocardial infarction and ischemic stroke. This is a well written review, providing an overview of current knowledge and highlighting conflicting evidence when existing. 

Comments to the manuscript:

1) No in-text reference to Figure 1 is included. Regarding this Figure, I wonder whether it would be useful to split into two, having Factor XIII structure (left panel of Figure 1) as separate figure and including at an earlier section, eg section 3.

2) A table summarizing the main findings of some of the key studies on Val34Leu FXIII polymorphism in myocardial infarction would be useful. This could also include relevant information on whether the study was performed in young or older individuals, on the effect of Val34Leu as a risk for MI or ischemic heart disease after MI. Given the controversies regarding Val34Leu polymorphism, such a table would be provide a helpful overview of current findings and would make this section a lot clearer to the reader.

3) It would be useful to the reader if the authors could expand a bit on the diagnostic tools and considerations on FXIII activity and MI. How do the authors envision that this will be achieved, what considerations should be kept in mind and how could these be applied in future clinical studies.  

4) In the conclusions section, the therapeutic potential of clinically available fibrogammin is mentioned, however this is not described anywhere in the main manuscript. Has this not been assessed in the context of MI or SI?

5) Line 310: ‘Chinese date even indicate data indicate that’, please correct.

6) Line 448: ‘rt-PA’, please include the full name description of this abbreviation

Comments on the Quality of English Language

English language is fine, minor editing is required 

Author Response

The manuscript by Jan Traub and co-authors provides a comprehensive summary of the current understanding on the role of Factor XIII in acute myocardial infarction and ischemic stroke. This is based on information generated by clinical studies, animal models and genetic polymorphism analysis, which overall have revealed a differential role of Factor XIII in acute myocardial infarction and ischemic stroke. This is a well written review, providing an overview of current knowledge and highlighting conflicting evidence when existing.

             Thanks for this summary and the positive feedback.

Comments to the manuscript:

1) No in-text reference to Figure 1 is included. Regarding this Figure, I wonder whether it would be useful to split into two, having Factor XIII structure (left panel of Figure 1) as separate figure and including at an earlier section, eg section 3.

This is a good comment. We included an in-text reference of Figure 1 in section 3 on page 2, where the structure of factor XIII is explained. A second in-text reference was added to the conclusions on page 13. We decided not to split the figure, as the combination of structure and significance of altered FXIII serum levels together may serve as graphical abstract or summary.

2) A table summarizing the main findings of some of the key studies on Val34Leu FXIII polymorphism in myocardial infarction would be useful. This could also include relevant information on whether the study was performed in young or older individuals, on the effect of Val34Leu as a risk for MI or ischemic heart disease after MI. Given the controversies regarding Val34Leu polymorphism, such a table would be provide a helpful overview of current findings and would make this section a lot clearer to the reader.

Thanks for this suggestion. Indeed, the paragraph on Val34Leu in myocardial infarction is quite extensive due to the amount of identified studies. We discussed this issue among all authors and decided to shorten this paragraph, as the role of Val34Leu in myocardial infarction (especially the risk of myocardial infarction) has been discussed elsewhere in form of meta analyses or review articles. This is why we would like to go without a table on this genetic issue, which is not the main focus of this review. Instead, the paragraphs on studies assessing the risk of MI in the context of Val34Leu were now replaced by:

“Briefly, there are meta-analyses concluding that FXIII Val34Leu polymorphism is associated with increased risk for coronary artery disease, especially MI, but not with coronary artery disease without MI [65,66]. In contrast, other meta-analyses suggested that FXIIIA Val34Leu polymorphism is a protective factor for MI generally [67] and in caucasians [68], which was also concluded by recent review articles on this controversial issue [69,70]. Two comprehensive meta-analyses have suggested that being a carrier of the Val34Leu allele and FXIII heterozygotes themselves are associated with a protective effect against premature MI [71]. Another meta-analysis claimed that the protective effect of the FXIII-A Val34Leu polymorphism could be significantly influ-enced by interactions between genes and environmental factors [72].”

3) It would be useful to the reader if the authors could expand a bit on the diagnostic tools and considerations on FXIII activity and MI. How do the authors envision that this will be achieved, what considerations should be kept in mind and how could these be applied in future clinical studies. 

In fact, this is a very important issue, which was kept quite short in the originally submitted work. Thus, the following paragraph was added to the section “6.7 Therapeutic considerations” after myocardial infarction:

“It should be noted that the optimal threshold for defining post-MI acquired FXIII deficiency remains unclear. There is currently a lack of comprehensive observational studies measuring plasma FXIII activity shortly after MI and linking it to outcomes such as ischemic heart failure, re-hospitalization, or mortality. It is worth mentioning that FXIII activity assays are readily available in clinical practice, which could stream-line the collection of this crucial data. Subsequently, initial human pilot studies or case reports can help assess the safety and appropriate dosage of intravenous FXIII re-placement therapy. The existing use of FXIII concentrates (i.e. fibrogammin) in routine clinical settings (e.g., before surgery in patients with FXIII deficiency) provides a sig-nificant advantage in this context. Finally, randomized clinical trials will be necessary to determine if early supplementation with FXIII after MI could serve as a viable therapeutic option for patients with (acquired) FXIII deficiency”

4) In the conclusions section, the therapeutic potential of clinically available fibrogammin is mentioned, however this is not described anywhere in the main manuscript. Has this not been assessed in the context of MI or SI?

Indeed, to the best of our knowledge, the potentially therapeutic effect of fibrogammin has not been assessed so far. As discussed above (3), we have now expanded the paragraph on therapeutic options after MI. Moreover, we added the following sentence to the conclusions section on page 13:

“(…) towards therapeutic applications of clinically already available FXIII (i.e. fibrogam-min®), which has not been assessed in this context so far.”

5) Line 310: ‘Chinese date even indicate data indicate that’, please correct.

Thanks for this remark, we apologize for the mistake. Following the suggestions of the editorial staff, we needed to re-phrase this sentence anyway:

“Data from Chinese studies suggest that the FXIIIA Leu34 allele may play a role in reducing the risk of developing MI.”

6) Line 448: ‘rt-PA’, please include the full name description of this abbreviation

             We replaced “rt-PA” by “recombinant tissue plasminogen activator”:

“Recombinant tissue plasminogen activator, which is intended to target fibrin, leads to a reduction in circulating fibrinogen levels and is closely associated with a decrease in plasminogen.”
